

# Revisiting Mirror Modes in the Plasma Environment of Comet 67P/Churyumov-Gerasimenko

Ariel Tello Fallau[1,2,*], Charlotte Goetz[3,4,*], Cyril Simon Wedlund[5], Martin Volwerk[5], and Anja Moeslinger[6]

[1]Department of Physics, University of Chile, Beauchef 850, Santiago, Chile
[2]Department of Mechanical Engineering, University of Chile, Beauchef 851, Santiago, Chile
[3]Department of Mathematics, Physics and Electrical Engineering, Northumbria University, Newcastle-upon-Tyne, United Kingdom
[4]ESTEC, European Space Agency, Keplerlaan 1, 2201AZ Noordwijk, The Netherlands
[5]Space Research Institute, Austrian Academy of Sciences, Schmiedlstraße 6, 8042 Graz, Austria
[6]Swedish Institute of Space Physics, 981 28 Kiruna, Sweden
[*]These authors contributed equally to this work.

**Correspondence:** ariel.tello@ing.uchile.cl

**Abstract.** The plasma environment of comet 67P provides a unique laboratory to study plasma phenomena in the interplanetary medium. There, waves are generated which help the plasma relax back to stability through wave-particle interactions, transferring energy from the wave to the particles and vice-versa. In this study, we focus on mirror mode structures (low-frequency, transverse, compressional and quasi-linearly polarised waves). They are present virtually everywhere in the solar system as long as there is a large temperature anisotropy and a high plasma beta. Previous studies have reported the existence of mirror modes at 67P but no further systematic investigation has so far been done. This study aims to characterise the occurrence of mirror modes in this environment and identify possible generation mechanisms through well-studied previous methods. Specifically, we make use of the magnetic field-only method, implementing a *B*–*n* anti-correlation and a new peak/dip identification method. We investigate the magnetic field measured by Rosetta from November 2014 to February 2016 and find 565 mirror mode signatures. Mirror modes were mostly found as single events, with only one mirror mode-like train in our dataset. Also, the occurrence rate was compared with respect to the gas production rates, cometocentric distance and magnetic field strength leading to a non-conclusive relation between these quantities. The lack of mirror mode wave trains may mean that mirror modes somehow diffuse and/or are overshadowed by the large-scale turbulence in the inner coma. The detected mirror modes are likely highly evolved as they were probably generated upstream of the observation point and have traversed a highly complex and turbulent plasma to reach their detection point. The plasma environment of comets behaves differently compared to planets and other objects in the solar system. Thus, knowing how mirror modes behave at comets could lead us to a more unified model for mirror modes in space plasmas.



# 1 Introduction

ESA's Rosetta mission was the first to rendezvous with a comet, the first to deploy a lander to a comet's surface (Taylor et al.,
2017), and the first to follow a comet in its orbit around the Sun. By studying the dust and gas from and the structure of the
nucleus and organic materials associated with the comet, via both remote and in situ observations, the Rosetta mission helped
unlock the history and evolution of our solar system. It also provided observations of the solar wind plasma interacting with
the cometary plasma. Within this plasma environment, we are given the unique opportunity to study an induced magnetosphere
at a small solar system object and explore how plasmas of different composition, momentum and energy interact (Goetz et al.,
2022).

One of the processes of interest in any plasma environment is the formation and propagation of waves. Different types of
waves and wave-like structures can be found in a cometary plasma environment (Ip, 2004; Goetz et al., 2022), e.g. magnetosonic
waves, Alfvén waves and ion cyclotron waves. Here, we focus on mirror modes, which have previously been found at comets
67P/Churyumov-Gerasimenko (67P, Volwerk et al., 2016a) and 1P/Halley (Mazelle et al., 1991; Schmid et al., 2014).

Mirror modes are low-frequency, long-wavelength, transverse, compressional and quasi-linearly polarised wave-like struc-
tures, which are non-propagating in the plasma rest frame.

They are present virtually everywhere in the solar system (planetary and cometary magnetosheaths, solar wind as magnetic
holes, etc.) as long as there is a temperature or pressure anisotropy (see Gary et al., 1993; Tsurutani et al., 2011). Specifically,
they contribute to the reduction of the temperature anisotropy by reducing the perpendicular temperature and redistributing en-
ergy in the plasma, hence making the magnetosheath globally stable to the generation of local temperature-driven instabilities.
Their occurrence can therefore also be used to infer plasma temperature anisotropies in their generation region.

In spacecraft observations, they usually appear as trains of magnetic field dips or peaks (or both) which are signatures of
magnetic bottles trapping high-density pockets of plasma, with the magnetic field magnitude $|B|$ and the plasma density $n$ in
antiphase, lasting from few seconds to a few tens of seconds. Although they are non-propagating in the plasma rest frame,
they can drift with the ambient plasma they are embedded in; hence the plasma parameters in the region the mirror modes are
generated in may differ from the region they are detected at. The instability at their origin arises from a temperature anisotropy
and preferential heating of the plasma along the perpendicular direction to the magnetic field. The drift mirror mode instability
criterion (MMI) can be written as follows (Hasegawa, 1969):

$$\text{MMI} = 1 + \sum_i \beta_{i\perp}\left(1 - \frac{T_{i\perp}}{T_{i\parallel}}\right) < 0, \tag{1}$$

where the sum is on all the species (ions and electrons) present in the plasma. The notations $\perp$ and $\parallel$ indicate the directions per-
pendicular and parallel to the background magnetic field. $T_{i\parallel}$ and $T_{i\perp}$ represent the species' parallel and perpendicular tempera-
tures, these temperatures define the parallel and perpendicular plasma beta as $\beta_{i\parallel} = 2\mu_0 N_i k_B T_{i\parallel}/|B|^2$ and $\beta_{i\perp} = 2\mu_0 N_i k_B T_{i\perp}/|B|^2$
respectively.

Mirror modes and Alfvén (left-hand polarised) ion cyclotron waves, both excited for large temperature anisotropies ($T_\perp >
T_\parallel$), are co-generated in the plasma. Mirror modes are found in weakly magnetised plasmas (high plasma beta $\beta \gg 1$), whereas



Alfvén ion cyclotron waves are found in low plasma $\beta$ conditions (Gary, 1992). Moreover, a minute addition of heavier, large-anisotropy ions ($He^{2+}$, $O^+$, etc.) to the solar wind has the effect in the wave dispersion relation of dampening the Alfvén ion cyclotron mode in favour of the mirror mode (Price et al., 1986), an effect that is expected to routinely occur in the cometary coma, where heavy ions are slowly incorporated into the plasma flow through solar wind mass loading and charge exchange
(Szegö et al., 2000; Simon Wedlund et al., 2019).

At comets and planets with an extended exosphere like Mars and Venus, the build-up of a temperature anisotropy is expected to occur in various ways and locations (see Simon Wedlund et al., 2022, and references therein): everywhere around the object (in the upstream solar wind or in the magnetosheath) through pick-up ion effects (Wu and Davidson, 1972), in the wake of the quasi-perpendicular shock (as at Earth and for any sufficiently developed bow shock), or close to the induced magnetospheric
boundary/magnetic pile-up boundary (IMB/MPB) by field-line draping and conservation of the first adiabatic invariant (if $|B|$ slowly increases due to compression of the field lines at the boundary, the conservation of the magnetic moment $\mu = mv_\perp^2/2|B|$ implies an increase of the perpendicular energy and thus temperature).

Since the first in-situ encounters with comets such as that of the ESA/Giotto spacecraft with 1P/Halley, mirror mode-like structures have been detected in their magnetosheath, especially close to their MPB (Mazelle et al., 1989; Mazelle et al., 1991;
Glassmeier et al., 1993; Schmid et al., 2014; Volwerk et al., 2016a, and references therein). In particular, Glassmeier et al. (1993) found, in the upstream part of the IMB, compressive, linearly polarised waves consistent with mirror modes ($|B|$ and electron density $n_e$ in antiphase) whereas the downstream part of the IMB contained fast-mode type magnetosonic waves. After revisiting the 1P/Halley datasets, Schmid et al. (2014) calculated the size of mirror mode structures to be of the order of 1–2 $H_2O^+$ gyroradii, suggesting that the main mechanism at the origin of the anisotropy thus generating mirror modes was pick-up
ion effects, rather than originating from the wake of the weak cometary shock.

At 67P, mirror mode structures were originally detected on a couple of days in 2015 (Volwerk et al., 2016a). Two different kinds of mirror mode-like structures were observed on 6 and 7 June 2015: one of small size generated by locally ionized water and one of large size generated by ionization and pick-up farther away from the comet.

The observed variations in the solar wind parameters, such as directional changes and increase in dynamic pressure, in both
solar wind propagation models used by Volwerk et al. (2016a), led to several interesting phenomena such as current densities in the current sheet of tens of $\mu A\ m^{-2}$ or several $nA\ m^{-2}$ (depending on the assumption of how fast Rosetta crosses this structure), evidence in the pile-up region for mirror mode-like structures generated by the newly created ions (with a size between one and three water-ion gyroradii as in Schmid et al., 2014, for 1P/Halley), and clear signatures of mirror mode-like structures outside the pile-up region (with a much larger size of 10 to 16 water-ion gyroradii).
As the plasma density data resolution is too low to check the pressure balance of the mirror mode structures, the magnetic-field-only method described in Lucek et al. (1999) was used by Volwerk et al. (2016a) to investigate the presence in the data of mirror modes. They used a Minimum Variance Analysis (MVA) of the RPC-MAG data over a sliding window to obtain the angles of the minimum and maximum variance directions with respect to a low-pass filtered (no longer than 10 min) background magnetic field. In order to identify mirror modes, the structures had to fulfil the following criteria: $\theta \geq 80$,
$\phi \leq 20$ and $\Delta B/B \geq 0.5$ (these quantities are also used in our current study and are specified in the method section). However,

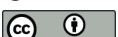



structures such as foreshock waves or fast-mode waves linked to pick-up ions are compressional in nature and may also fulfil these criteria. In those cases, plasma measurements are necessary to lift the ambiguity; specifically, by checking for an anti-correlation between the plasma density and the magnetic field magnitude that is expected for mirror-mode structures (Hasegawa, 1969).

Linear magnetic holes have been suggested to have an association with mirror modes because the plasma in and close to the magnetic hole is often only marginally stable with respect to the mirror mode instability (Winterhalter et al., 1995; Neugebauer et al., 2001; Stevens and Kasper, 2007). The fact that the magnetic holes are isolated structures, and not part of a periodic chain of structures as one would expect from mirror modes, would suggest that linear magnetic holes are remnants of mirror modes; these linear magnetic holes are possibly the result of coalescing mirror mode structures (Winterhalter et al., 1995; Zhang et al., 2009) and possibly propagating as soliton structures through a medium that is mirror mode stable (Baumgärtel, 1999; Sperveslage et al., 2000).

In parallel with mirror mode detections, magnetic holes in the modified solar wind at comet 67P were detected in the spring of 2015 when, for the first time, they were found in a cometary environment even when solar wind protons are almost absent (Plaschke et al., 2018). This shows that solar wind structures can convect deep into the plasma environment with only marginal modification, even when solar wind ions have been deflected and substituted with accelerated cometary ions.

The main objective of this study is to investigate when and where mirror modes can be found near the comet and how their occurrence and morphology depend on the plasma parameters. This should allow to infer how the mirror modes are generated and how this generation mechanism compares to other comets and planets.

In the following, we first describe the method used to identify mirror modes in the plasma at 67P. Then, we present two case studies of mirror modes and a statistical study of their occurrence and properties. We end with a discussion of the results and a brief conclusion.

## 2 Methods

### 2.1 Instrumentation

In this study, we use the observations by the Rosetta Plasma Consortium (RPC), specifically three subunits: the magnetometer (RPC-MAG, Glassmeier et al., 2007a), the Langmuir Probe (RPC-LAP, Eriksson et al., 2007), and the Mutual Impedance Probe (RPC-MIP, Trotignon et al., 2007). As all of the instruments have different capabilities and limitations, we will discuss them briefly in the following.

The RPC-MAG magnetometer consists of two fluxgate sensors, which are mounted on a boom and can measure the magnetic field components at a frequency of up to 20 vectors per second. As we are interested in structures of time scales of a few seconds to several tens of seconds, we downsample the magnetic field data to 1 Hz[1] to optimise data volume. This also has the advantage of eliminating the influence of the spacecraft reaction wheels, which introduce a signature that is only found above

---

[1]https://archives.esac.esa.int/psa/#!Table%20View/RPC=instrument, Dataset Identifier Version V9.0





those frequencies (Glassmeier et al., 2007b). The major caveat when using RPC-MAG data is the offset uncertainty. Even after extensive calibration efforts a systematic offset of about $\pm 3\,\text{nT}$ remains per component. Low field values and especially the angles calculated from those values should therefore be treated very carefully and ideally eliminated from the study. Due to this constraint and the availability of plasma density data, we only use data from November 2014 to February 2016 for this study.

The plasma density was measured by the Langmuir Probe (LAP, Eriksson et al., 2007) and by the Mutual Impedance Probe (MIP, Trotignon et al., 2007). The LAP instrument utilises a pair of spherical Langmuir probes for measurements of basic plasma parameters such as ion and electron currents, which depending on instrument mode can be converted into density, spacecraft potential, electric field and others. MIP can provide electron densities and temperatures from mutual impedance spectra. For our study, we use two different estimates of the density: the sweep-derived densities and the density derived from the probe current. The first is only available at a time resolution of minutes, which is useful for context, but not enough to resolve the details of the mirror mode structures. For the second, the dataset is derived from LAP currents in combination with MIP density measurements. This combination is the only way to produce accurate, reliable and high time resolution data for the plasma density (Breuillard et al., 2019). Some of these cross-calibrated data values are negative, which is due to the instrument's calibration threshold. These intervals are omitted from the analysis. In order to compare to the magnetic field, the density values are interpolated to the same time stamps as the magnetic field data.

The neutral gas density was measured by the Rosetta Orbiter Spectrometer for Ion and Neutral Analysis ROSINA-COPS (Balsiger et al., 2007). ROSINA consists of two mass spectrometers for neutrals and primary ions with complementary capabilities and a pressure sensor. The neutral gas density is derived from the pressure sensor measurements. To derive an estimate of the local gas production rate, we use a neutral outgassing velocity of $1\,\text{km s}^{-1}$, a simple spherically symmetric model Haser (1957) and the measured neutral gas density.

The spacecraft and comet ephemerides were derived using freely available SPICE kernels[2]. If not stated otherwise, all coordinates are given in the CSEQ system, where the comet is at the origin, the $x$-axis points towards the Sun and the $z$-axis is parallel to the Sun's North pole direction. The $y$-axis completes the right-handed coordinate system.

## 2.2 Selection method

Ideally, the entire Rosetta dataset at the comet should be used to search for mirror modes; however, due to data availability and calibration issues, we are constrained to only using the period from November 2014 to February 2016. This still gives quite good coverage, only very low activity cannot be investigated. To find signatures of mirror modes in the RPC data, we proceed in two steps. First, we adapt and use the magnetic field-only detection method described in Volwerk et al. (2016a) and further refined in Simon Wedlund et al. (2022) to identify mirror mode candidates in the data. Second, we refine our initial event selection by searching for an anti-correlation between magnetic field and density measurements. Then, for the anti-correlated events, we ensure a wave-like behaviour by only retaining events that exhibit a clear minimum or maximum in the magnetic field strength and plasma density data.

---

[2]https://www.cosmos.esa.int/web/spice/spice-for-rosetta





**Magnetic field-only method:**

First, as our structures are of the order of a couple of minutes, we calculate a 10-minute moving mean of the magnetic field to estimate the background field vector $\mathbf{B}_{\text{bg}}$. Second, we perform a minimum variance analysis (MVA) on the magnetic field vector to determine the direction of minimum and maximum variance (Sonnerup and Scheible, 1998). This is done for a 30 s moving window for each second of data, as in Volwerk et al. (2016a). Then, the angles between the minimum and maximum variance directions ($\mathbf{b_{min}}$ and $\mathbf{b_{max}}$) with respect to that of the background magnetic field vector ($\mathbf{B}_{\text{bg}}$) are calculated. Following Simon Wedlund et al. (2022), $\theta$ denotes the angle $\angle\left(\mathbf{b_{min}}, \mathbf{B}_{\text{bg}}\right)$ and $\phi$ denotes the angle $\angle\left(\mathbf{b_{max}}, \mathbf{B}_{\text{bg}}\right)$. Mirror modes are compressional and linearly polarised, therefore we require that $\theta$ be small, while $\phi$ should be large. The exact values of the thresholds and references on which they are based can be found in Table 1. Furthermore, as in Simon Wedlund et al. (2022); we use the maximum, intermediate and minimum eigenvalues to calculate the eigenvalue ratios $\lambda_{\text{max}}/\lambda_{\text{int}}$ and $\lambda_{\text{int}}/\lambda_{\text{min}}$ to ensure that the wave-like modes are quasi-linearly polarised, that the maximum variance direction (the tangential component of the eigenvector triad) is well defined, and that the quasi-degeneracy of the covariance matrix is kept to the two minimum eigenvalues (Criteria 3 and 4 of Table 1).

Mirror modes are highly compressive, therefore we also require that the magnetic field variations (defined at eq. 2) are large (Criterion 1 in Table 1).

$$\Delta B/B = \left|(B - B_{\text{bg}})/B_{\text{bg}}\right| \tag{2}$$

Here, $B$ is the magnetic field magnitude and $B_{\text{bg}}$ is the background magnetic field magnitude.

| # | Criterion | Value | Reason | Example reference(s) |
|---|---|---|---|---|
| 1 | $\Delta B/B$ | $\geq 0.5$ | Compressional structure | Génot et al. (2009a); Volwerk et al. (2016b) |
| 2 | $\phi_{\text{maxV}}$ | $\leq 20$ | Compressional structure | Génot et al. (2009b); Volwerk et al. (2016b) |
| | $\theta_{\text{minV}}$ | $\geq 70$ | Perpendicular wave propagation direction | Volwerk et al. (2016b); Simon Wedlund et al. (2022) |
| 3 | $\lambda_{\text{max}}/\lambda_{\text{int}}$ | $\geq 3$ | Quasi-linearly polarised wave | Génot et al. (2001); Soucek et al. (2008) |
| 4 | $\lambda_{\text{int}}/\lambda_{\text{min}}$ | $\leq 6$ | Quasi-linearly polarised wave | Génot et al. (2001); Soucek et al. (2008) |
| 5 | $B$–$n$ anti-correlation | $\leq -0.7$ | Pressure equilibrium structures | Volwerk (2016); Simon Wedlund et al. (2022) |

**Table 1.** Mirror mode selection criteria ensuring that the detected structures are compressional, linearly polarised, and have their magnetic field in antiphase with plasma density measurements.

The magnetic field-only method returns values of $\phi$, $\theta$, $\lambda_{max}$, $\lambda_{min}$ and the maximum $\Delta B/B$ of the interval for each second of the day. Then, we apply all selection criteria shown in Table 1 for each second of data. If the criteria are satisfied and if a mirror mode candidate has a time difference equal to or lower than 15 s with the next event, then both events are combined and considered one event (as in Volwerk et al., 2016b).

After this selection, we obtain a list of 32026 mirror mode candidates throughout the mission between November 1st 2014 and February 29th 2016. A closer inspection reveals that many of the mirror mode candidates found by the magnetic field only method are compressional magnetic field structures, also known as steepened waves (Ostaszewski et al., 2020). This is





expected because compressional structures also have a large $\Delta B/B$ and may satisfy the angle criteria as well. However, it is clear that they are not mirror modes and lack the characteristic $B$–$n$ antiphase behaviour expected for mirror modes; therefore the method needs to be expanded to eliminate those false positive detections.

We wanted to validate the magnetic field-only method through the events found by Volwerk et al. (2016a) on 6 and 7 June 2015. In this study, there are 2 relevant time intervals, 6 June (19:10-19:12 and 22:31-22:33) and the beginning of 7 June (01:10-03:40). We only found structures on 6 June from 22:31 to 22:32; this may be because of the data version of the magnetometer dataset available at the time. Since then, a new version with improved offset and temperature calibration was published. With the new offset calibration, the events found by Volwerk et al. (2016a) are no longer identified as mirror modes.

**Anti-correlation method:** While many compressional structures are characterised by the magnetic field and plasma density being in phase, mirror modes show a clear anti-correlation between those two quantities. Thus, a new selection criterion is introduced, which formalises the method described in Simon Wedlund et al. (2022) (see also Table 1, Criterion 5). We compute the Pearson correlation coefficient $\mathcal{R}$ between the magnetic field and the density for each mirror mode candidate interval. To ensure that high-frequency variations do not interfere with this calculation, the data are smoothed with a moving average over 3 s, which allows for the removal of most high frequencies while preserving the general trend. We only retain those candidates where the correlation coefficient is lower than $-0.7$, which means that the least-squares linear regression model can explain at least $\sim 50\%$ of the variation in the magnetic field data.

After applying the $B$–$n$ anti-correlation criterion on the initial mirror-mode candidate database, only 2508 possible events remain. Of those events, there are cases where density and magnetic field anti-correlate but the signature does not show a clear wave-like train, i.e. at least one peak or dip in the interval in question (Fig. A1). Thus, to filter out these mirror mode candidates, a peak/dip identification method is also implemented.

**Peak/dip identification method:** The peak/dip identification method consists of three steps. Firstly, to filter out the mirror mode candidates that have no peaks or dips, a linear fit is calculated (separately) for the magnetic field strength and for the plasma density data during the mirror mode candidate time interval. If the linear fit's $\mathcal{R}^2$ value is higher than 0.7 the candidate is removed due to its linear behaviour. Secondly, for the rest of the candidates that satisfy the $\mathcal{R}^2$ condition, the peaks and dips with the highest prominence (height above the background) for both magnetic field strength and plasma density are computed. Finally, if the time difference between the magnetic field strength minimum and plasma density maximum (and vice versa) is lower than half of the maximum prominence (of the plasma density and magnetic field strength peaks-dips) in the time interval, then we can ensure that these minima and maxima are related in time to each other and that we retain an anti-correlated, mirror mode-like behaviour. The method is illustrated in Appendix A and Fig. A2. In the end, after applying these successive steps to ensure that only unambiguous highly compressional mirror modes are retained, 565 'true' mirror mode-like events remain.

It should be noted that ion cyclotron waves have not been detected at comet 67P/Churyumov-Gerasimenko, possibly due to the elevated plasma-$\beta$ within the inner coma (Götz, 2019). Also, the presence of magnetic holes at 67P (Plaschke et al., 2018) may suggest a high plasma-$\beta$ environment (Baumgärtel, 1999), more conducive to the generation of mirror modes. Thus, although we are unable to check if these events are indeed associated with a temperature anisotropy due to the unavailability of data, for readability, we will refer to these events as mirror modes in the following discussions.





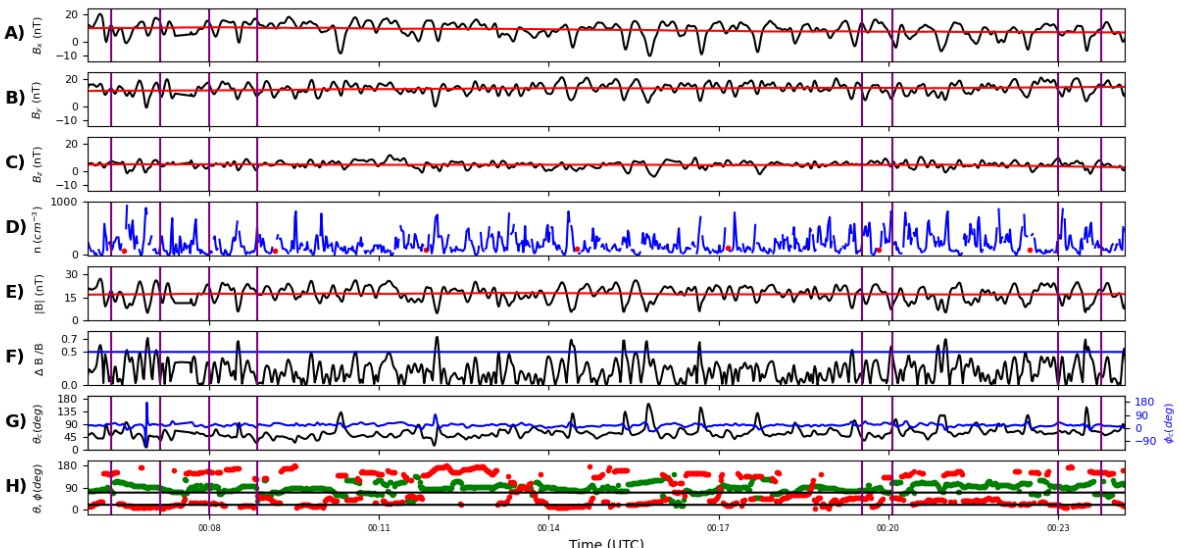

**Figure 1.** Mirror mode train found on 10 February 2015. Four unambiguous events (marked by purple lines) were detected from 00:06:16 to 00:07:08, 00:08:08 to 00:08:51, 00:19:32 to 00:20:04 and 00:23:00 to 00:23:46 respectively. **(a-c)** Magnetic field components (in CSEQ) in black and the rolling average of the data in red. **(d)** Plasma density, LAP sweep derived density in red, and MIP-LAP cross calibrated density in blue. **(e)** Magnetic field strength in black and the rolling average in red. **(f)** $\Delta B/B$ from the mirror mode identification procedure. **(g)** Magnetic field cone and clock angles in black and blue respectively. **(h)** Angles $\theta$ (green) and $\phi$ (red) between the background field directions and the minimum/maximum variance directions from the mirror mode identification procedure respectively.

## 3 Results

First, we will present two individual cases for mirror modes, in order to verify our selection method and dive deeper into the structure of the mirror modes themselves. Second, we will perform a statistical study of the occurrence of mirror modes in the plasma environment. This will allow us to make inferences about where they could be generated and the plasma through which they travel.

### 3.1 Case Studies

In the following section we have chosen to focus on two particular events: the first one is an example of a train of mirror modes that are similar to the classical structures found in the environment of comet 1P/Halley (Volwerk et al., 2014), at 67P (Volwerk et al., 2016a) in a previous study and at Venus and Mars (Volwerk et al., 2016b; Schmid et al., 2014; Simon Wedlund et al., 2022). The second event is representative of a set of mirror modes that are found deeper in the coma of the comet.





**Event 1: 10 February 2015**

Fig. 1 shows about 15 minutes of observations in February 2015, when the spacecraft was at a radial distance of 105 km with respect to the comet, and the gas production rate was about $4.0 \times 10^{26}$ s$^{-1}$ as derived from measured neutral densities. The average plasma density and average magnetic field strength of this event are 200 cm$^{-3}$ and 17 nT respectively.

   Four mirror mode structures (marked by purple lines) were identified by the selection method described in Sect. 2.2; each event always contains at least one peak in $\Delta B/B$ (panel f) that reaches above 0.5, according to our selection method. In addition,

all events exhibit the necessary anti-correlation of the magnetic field magnitude (panel e) and the density (panel d) as well as the requisite angle constraints (panel h).

   Although only four intervals were identified as unambiguous mirror mode structures, there are other signatures in the magnetic field, especially between 00:14 and 00:17 UT, that resemble the other mirror mode events. We suspect that these do not fulfil our stringent criteria but could still be mirror modes, following similar remarks made in Simon Wedlund et al. (2022).

This is the only interval in our database where such a train of mirror modes was observed.

   It should be noted that the third mirror-mode interval in Fig. 1 (from left to right) only satisfies the $\Delta B/B$ criterion at the end of the interval instead of where there is an increase in the magnetic field strength and a decrease in the plasma density; this is due to the MVA time interval used to obtain the $\Delta B/B$ values.

   In accordance with Volwerk et al. (2016a) we compute the eigenvectors for an interval of 30 s each, and assign the highest

value of $\Delta B/B$ within those 30 s to that interval. Then, although there is no peak in the $\Delta B/B$ criterion where the magnetic field strength and plasma density are anti-correlated, there is a $\Delta B/B$ peak that satisfies the criterion somewhere in the 30 s interval.

   On the other hand, for the third event in Fig. 1, the selection method did not identify the second of data at the time of the peak in $\Delta B/B$ as a mirror mode because, although that second satisfies the $\Delta B/B$ and there is a respective $B$–$n$ anti-correlation, it does not satisfy the eigenvalue or angle criteria.

We also computed the magnetic field angles in the CSEQ system as an extra check for linear polarisation, as shown in the paper of Tsurutani et al. (2011), where they found that the angles were typically below 10°. The cone angle $\theta_c$ and the clock angle $\phi_c$ are defined as:

$$\theta_c = \mathrm{atan}\left(\frac{\sqrt{B_y^2 + B_z^2}}{B_x}\right) \tag{3}$$

$$\phi_c = \mathrm{atan}\left(\frac{B_z}{B_y}\right). \tag{4}$$

As can be seen in Table 2, the variation of these two angles is small but slightly larger than found by Tsurutani et al. (2011). This is not unexpected, as the cometary environment is more turbulent and smaller than magnetosheaths at planets with intrinsic magnetic fields (e.g. Earth, Jupiter, Saturn) and mirror mode structures may not have had enough time to fully grow. In summary, this event is most like the classical picture of mirror modes as described in Volwerk et al. (2016a).





|  | Beginning | End | $\Delta\theta_c$ (°) | $\Delta\phi_c$ (°) |
|---|---|---|---|---|
| 1st Event | 00:06:16 | 00:07:08 | 20.7 | 19.9 |
| 2nd Event | 00:08:08 | 00:08:51 | 17.8 | 19.2 |
| 3rd Event | 00:19:32 | 00:20:04 | 21.0 | 11.3 |
| 4th Event | 00:23:00 | 00:23:46 | 17.3 | 14.6 |

**Table 2.** Dates and maximum variations for the cone and clock angles for the train of mirror mode events using a 10-minute moving standard deviation. The time length used for the moving standard deviation does not impact the angle variations significantly.

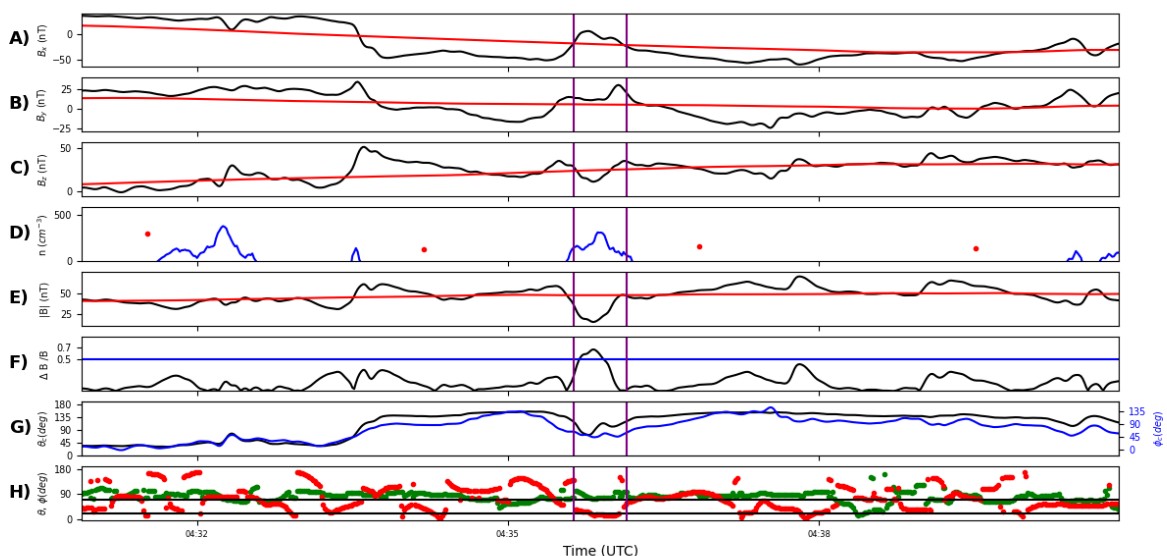

**Figure 2.** 17 August 2015 mirror mode single event. **(a-c)** Magnetic field components (in CSEQ) in black and the rolling average of the data is in red. **(d)** Plasma density, LAP sweep derived density in red and MIP-LAP cross calibrated density in blue (negative values are not shown in this subplot). **(e)** Magnetic field strength in black and the rolling average in red. **(f)** $\Delta B/B$ from the mirror mode identification procedure. **(g)** Magnetic field cone and clock angles in black and blue, respectively. **(h)** Angles $\theta$ (green) and $\phi$ (red) between the background field directions and the minimum/maximum variance directions from the mirror mode identification procedure, respectively. The mirror mode event is marked by purple lines lasting from 04:35:38 to 04:36:09.

**Event 2: 17 August 2015**

Event 2, shown in Fig. 2, occurred on 17 August 2015, when the spacecraft was at a radial distance of 327 km to the nucleus. The gas production rate based on in-situ measurements was $1.2 \times 10^{28}\,\mathrm{s^{-1}}$, at a heliocentric distance of 1.24 AU. The average plasma density and average magnetic field strength of this event are $168\,\mathrm{cm^{-3}}$ and $29\,\mathrm{nT}$ respectively.





In this case, we show an isolated event (marked by purple lines) identified by the selection method described in Sect. 2.2; representative of the events found around perihelion. Again, there is a clear anti-correlation between the density and the

magnetic field magnitude (panels d and e). $\Delta B/B$ is also clearly higher than 0.5, the minimum and maximum variance angles are above 70° and below 20°, as requested by the selection method. The cone and clock angle variations ($\Delta\theta_c = 46.1°$ and $\Delta\phi_c = 39.9°$ respectively) indicate that there is a slightly larger rotation of the field during the mirror mode event. Again, this is larger than found by Tsurutani et al. (2011) and larger than for the other event. We attribute this to the fact that this event was found at a high gas production rate in the innermost coma where the cometary environment is at its most turbulent.

**Comparison of Events**

The two examples show that, because our mirror mode identification criteria are quite stringent, we manage to find events that are particularly clear. Both events show structures with unambiguous mirror-mode characteristics, albeit at different times during the development of the plasma environment of the comet. Mirror modes are usually found to have sizes of roughly 1-3 water ion gyroradii ($r_g$) at both comet 1P/Halley and comet 67P if they are locally generated and at $10 - 16r_g$ if they are

generated upstream and convected to the point of observation (Schmid et al., 2014; Volwerk et al., 2016a). To estimate this characteristic, we introduce the normalised length scale $L^*$:

$$L^* = \frac{L}{r_g} \tag{5}$$

with $L = v_B \, \Delta t$, which becomes:

$$L^* = \frac{v_B \Delta t}{\frac{v_B m}{qB}} = \Delta t \, \omega_g \tag{6}$$

where $\Delta t$ is the duration of the mirror mode, $v_B$ is the ion bulk velocity and $\omega_g$ is the ion gyrofrequency $\omega_g = qB/m$. This calculation assumes that the mirror mode convects with the bulk flow and that the pick-up ions are gyrating around the plasma reference frame, with $v_{perp} = \|v_B\|$. We now assume that the mirror mode wave train in Event 1 is embedded in a 20 nT field. This results in $L^* = (5.5, 4.6, 3.4, 4.9)$ for the four events for water ions. The isolated mirror mode in Event 2 lasts 31 s. If we assume a pure water coma with a background magnetic field of 30 nT (see Fig. 2), this gives $L^* \sim 4.9$. All observed structures

are of roughly the same size with regards to the local pick-up ion gyroradius. The values are slightly larger than what was previously found for locally generated mirror modes, indicating that the mirror modes have likely been generated upstream of Rosetta's location and have convected downstream with the plasma flow. The lack of trains of mirror modes would then imply that mirror modes diffuse and/or are overshadowed by the large-scale turbulence in the inner coma (Goetz et al., 2016). This is reminiscent of the findings by Plaschke et al. (2018) who showed that magnetic holes from the solar wind can traverse the

coma and are modified by the changing plasma conditions.





## 3.2 Statistical Study

We have identified 565 mirror mode intervals in the Rosetta data from November 2014 to March 2016. The large number of events found with our method allows us to characterise the mirror modes and to study whether there are specific conditions under which the mirror modes occur preferentially. We therefore, perform a statistical study of the events.

There are two limiting factors to this statistical study: 1) the availability of well calibrated magnetic field data and 2) the availability of high time resolution density data. 1) means that we only cover gas production rates of roughly $Q > 10^{26}\,\mathrm{s}^{-1}$, but 2) is not as easily categorised. We therefore, normalise all detection rates to the number of available density observations, this normalisation consist of dividing the amount of mirror modes found per day with the amount of plasma density availability in seconds per day.

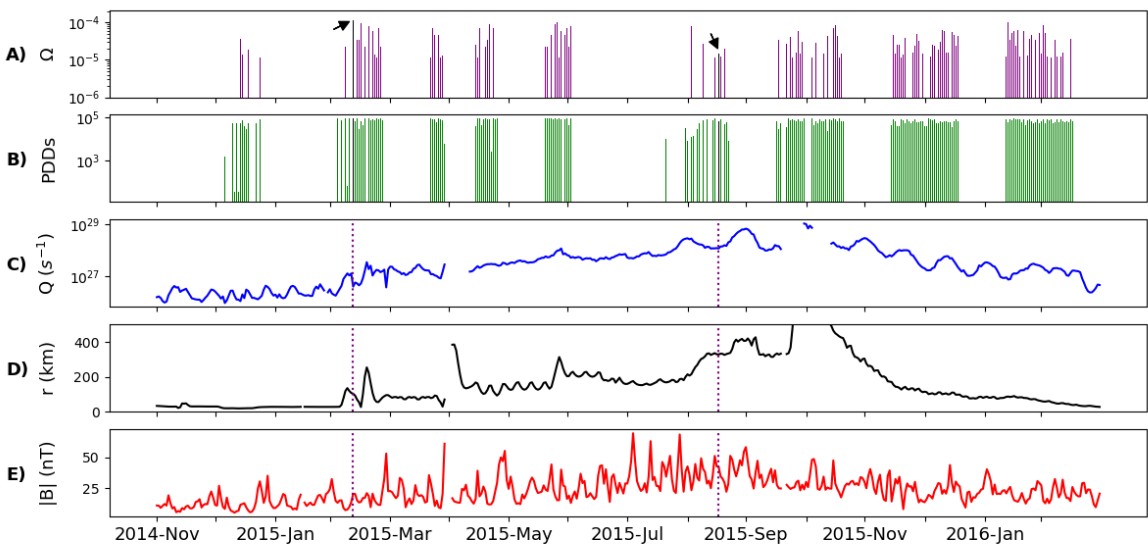

**Figure 3. (a)** Normalised mirror mode occurrence rate (number of mirror modes divided by the plasma density data in seconds per day). The left arrow indicates the train of mirror modes shown in Fig. 1 and the right arrow the perihelion event shown in Fig. 2. **(b)** Plasma density data in seconds (PDDs) per day. **(c)** Cometary gas production rate $Q$ in $\mathrm{s}^{-1}$. **(d)** Cometocentric distance $r$ of the spacecraft in km. The $y$ axis is range-limited for better visibility: this removes part of the dayside excursion that went up to 1500 km from the comet nucleus around October 2015. **(e)** Average magnetic field strength per day in nT. Vertical dotted purple lines are added in panels (c)-(e) in order to pinpoint the timing of the events shown in Figs. 1 and 2.

Fig. 3 explores the link between the found mirror modes with the gas production rate, cometocentric distance and magnetic field strength over time. The detection rate (panel a) is normalised to the availability of plasma density observations (panel b). The two events discussed in the previous section are marked with arrows or dotted vertical lines.



Overall, mirror modes are detected whenever plasma density data is available, independent of gas production rate, heliocentric distance or magnetic field strength. As expected, the outgassing rate and magnetic field magnitude increase as the comet

reaches perihelion (Hansen et al., 2016; Goetz et al., 2017), but there seems to be no pattern to the occurrence rate of the mirror modes that correlates with this increase. There is also no indication that mirror modes preferentially occur far from (or near) the nucleus. However, the spacecraft trajectory and outgassing rate are not independent parameters and therefore a more careful treatment is necessary.

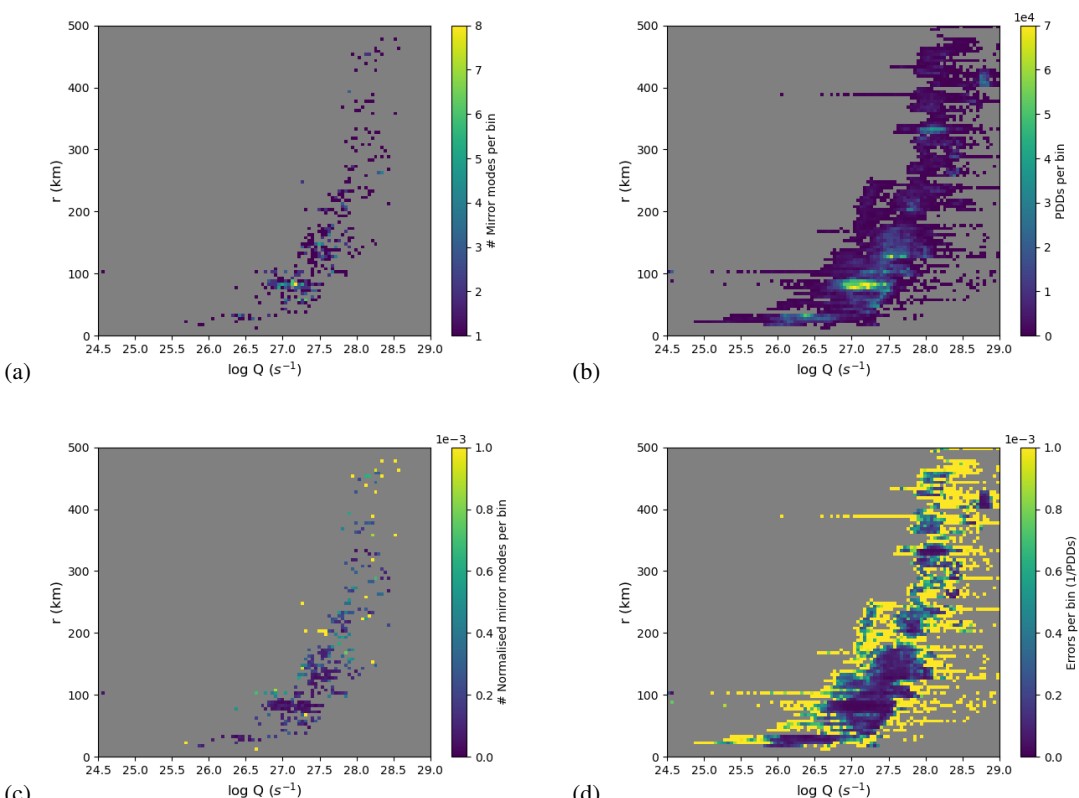

**Figure 4.** Cometocentric distance vs gas production rate 2D histograms using $100 \times 100$ bins. **(a)** Mirror modes' distribution (# mirror modes per bin). **(b)** Plasma density data availability (PDDs per bin). **(c)** Normalised mirror modes distribution (# mirror modes/ PDDs per bin). **(d)** One-count normalisation errors (for more see text).

The following histograms were plotted using the mean value of the cometocentric distance, magnetic field magnitude, $x$

coordinate and $\rho$ for each detected mirror mode (these values are obtained using data relative to each mirror mode time interval), where $\rho$ is defined as:

$$\rho = \sqrt{y^2 + z^2} \tag{7}$$





Bins of dimensions 100×100 were used for all histograms. Fig. 4 shows the mirror mode occurrence in a gas production rate and cometocentric distance histogram, with panel a showing where mirror modes are found, panel b the availability of data,

panel c the normalised mirror mode occurrence rate and panel d the normalisation errors assumed as ±1 mirror mode normalised by plasma density data availability per bin. As above, it is difficult to discern any trends. The mirror mode occurrence rate seems to be entirely determined by the coverage of the spacecraft. Due to operational constraints, the spacecraft was consistently at higher cometocentric distances for high gas production rates, and the closest approach distance was strongly correlated with the dust environment and therefore the outgassing rate. It stands to reason that Rosetta spent most of the mission in the same

region of the plasma environment (Goetz et al., 2016). As the gas production rate increases, so does the cometocentric distance of the spacecraft and the expansion of regions like, e.g., the diamagnetic cavity, solar wind ion cavity and the bow shock. It is therefore not surprising that there is little variation in the mirror mode occurrence rate, as Rosetta stayed roughly in the same region at all times. While there are some bins with high (yellow) occurrence rates, those all occur when there is poor coverage of the density and therefore they are associated with large error bars as shown in panel d. We conclude that there is

no discernible trend in the occurrence rate of the mirror modes.

The measured magnetic field strength is also not independent of the gas production rate (Goetz et al., 2016), therefore Fig. 5 shows the distribution of mirror modes in a $Q - B$ diagram. The format of panel a, b, c and d is the same as in Fig. 4. We again note that most bins with high occurrence rates are found in areas of poor density data coverage and we therefore have to take into account high error bars (panel d). As before, there is no discernible trend, mirror modes are found at all gas production

rates and magnetic fields.

Lastly, we investigate the spatial occurrence rate of these mirror modes. Fig. 6 shows their distribution in a $x - \rho$ diagram in cylindrical CSEQ coordinates.

Again, due to operational constraints, the coverage is quite poor. Nevertheless, we should be able to determine whether mirror modes occur preferentially at high $x$ or at any distance. As above, the mirror mode occurrence rate (panel c) does not

show any clear pattern in this coordinate system.

From these figures we conclude that there are no discernible factors, whether they are magnetic field strength, cometocentric distance or outgassing rate, driving the mirror mode occurrence rate.

The morphology of the mirror mode, specifically whether the magnetic field exhibits a peak (enhancement in the $B$-field magnitude) or a dip (depressions in $|B|$), has previously been related to the stage of mirror mode evolution. At Jupiter, Joy

et al. (2006), following similar comments at Saturn (Bavassano Cattaneo et al., 1998), remarked that dip structures occurred more often in the deep magnetosheath close to the magnetopause in a relatively low-beta plasma, whereas peaks occurred in the middle of the magnetosheath in a comparatively higher-beta plasma. Structures containing a mix of dips and peaks, labelled 'others' or 'quasi-periodic' in the study of Joy et al. (2006), were seen everywhere in the magnetosheath, but more specifically closer to the bow shock. This behaviour is also seen at Earth (Soucek et al., 2008). Hence, extending the model

of Bavassano Cattaneo et al. (1998), Joy et al. (2006) hypothesised an evolutionary link between these morphologies, with trains of mirror modes created first as a mix of peaks and dips, and progressively evolving towards peaks in the non-linear saturation phase of the instability, and, convected down to the magnetopause, finally decaying as dips. Although the first part





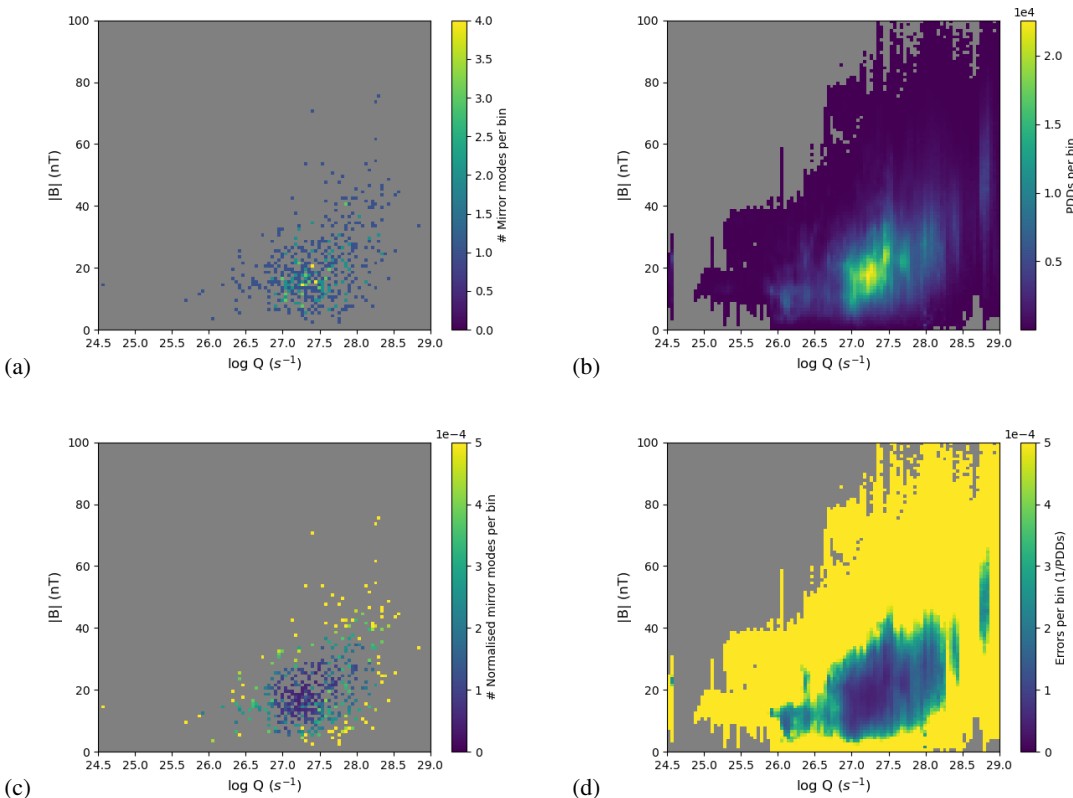

**Figure 5.** Magnetic field strength vs gas production rate 2D histograms using $100 \times 100$ bins. **(a)** Mirror modes distribution (# mirror modes per bin). **(b)** Plasma density data availability (PDDs per bin). **(c)** Normalised mirror modes distribution (# mirror modes/ PDDs per bin). **(d)** One-count normalisation errors (for more see text)

.

could be substantiated with in-situ Earth magnetosheath measurements, the decaying part of the scenario into magnetic dips remained difficult to prove at the time, with still more theoretical and modelling work actively done (Ahmadi et al., 2017).

Such an evolution between morphologies of trains of mirror modes is reminiscent of similar conclusions based on observations at comet 67P of often-related structures such as magnetic holes (Plaschke et al., 2018), with the magnetic structure sometimes changing to a more complex form than a simple dip.

We therefore determined for each event whether the change in magnetic field was an increase (peak) or decrease (dip) above the background field. This determination is not always clear, as some events exhibit both peaks and dips: we therefore also

introduced the category "both" to refer to them (see Fig. A3 in Appendix B).

In total, there are 150 peaks, 185 dips and 230 peaks and dips simultaneously ('both'). Fig. 7 shows the normalised occurrence rate of all three categories per month.





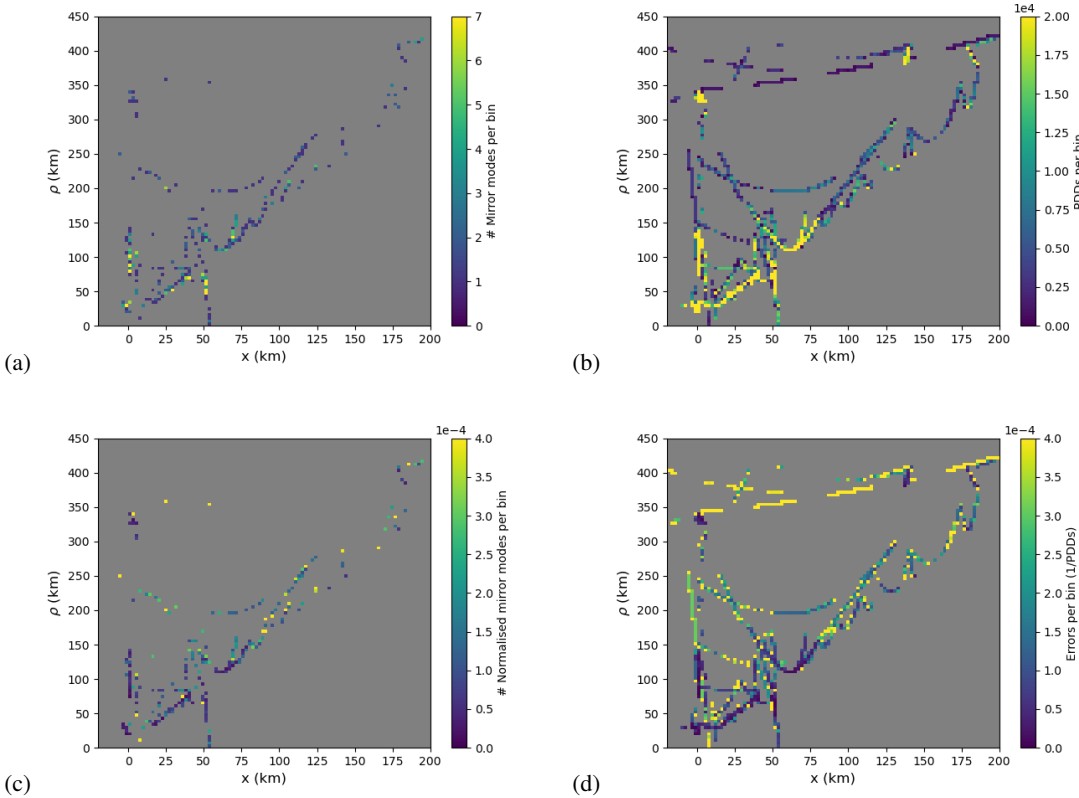

**Figure 6.** 2D histograms of $\rho$ vs $x$ using $100 \times 100$ bins. **(a)** Mirror modes distribution (# mirror modes per bin). **(b)** Plasma density data availability (PDDs per bin). **(c)** Normalised mirror modes distribution (# mirror modes/ PDDs per bin). **(d)** One count normalisation errors (for more see text).

No density data is available for January 2015 and very little data is available for July 2015. The number of peaks seems to rise steadily towards perihelion, with most being detected in June 2015. There is also a decrease in the number of observations

after perihelion. Specifically, the mean number of peaks dropped by 40% of the mean peak events before perihelion. No such clear trend is visible for dips, dropping 5% of the mean number of observations before perihelion. The mean number of both peaks and dips dropped by 15% of the value before perihelion.

In general, all detection rates decrease after 67P reaches perihelion, with peaks being the most sensitive to this trend. More specifically, with a slight increase towards November 2015 followed by a substantial decrease. The last category which includes

both peaks and dips behaves similarly to the peaks.

On Fig. 7, the Rosetta mission spanned different cometocentric and heliocentric distances as well as varying outgassing rates and solar EUV conditions, which are difficult to disentangle. Before and after perihelion conditions (13 August 2015), the outgassing rate of the comet differs in evolution and intensity (Hansen et al., 2016), which is expected to affect the efficiency of the different possible sources of temperature anisotropy (pick-up ion unstable distributions, quasi-perpendicular shock, etc.,





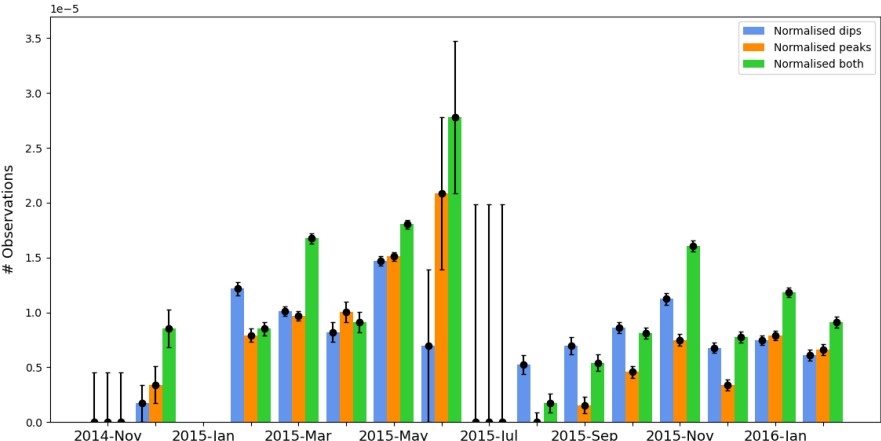

**Figure 7.** Normalised mirror modes with an increase (peak), decrease (dip) or both(category 'both') in the magnetic field magnitude above the background magnetic field (# mirror modes/ PDDs per month). Error bars show the one-count errors (±1 normalised mirror mode per month). January 2015 has no error bars because there is no plasma density data availability for this month. Also, although we did not identify any mirror modes in July 2015, this month has the largest error bars because it only contains a few days with plasma density data availability (see Fig. 3 panel b).

see Mazelle et al., 1989; Mazelle et al., 1991). Around perihelion, the standoff position of the well-formed cometary bow shock was estimated from realistic models to be about $15$–$20 \times 10^3$ km (Alho et al., 2021). At the time, Rosetta orbited around 300 km from the nucleus, i.e., at least 50 times closer than the expected shock location. After perihelion, the outgassing rate decreased steadily, with the shock's expected standoff distance reducing faster than Rosetta's cometocentric distance, as it stayed within 100 km of the nucleus. This indicates that, around perihelion, Rosetta was relatively deeper in the magnetosheath than later

on, with a sufficiently large magnetosheath for mirror-mode structures to evolve. The perihelion data in August 2015 shows no peaks which is compatible with the idea of Bavassano Cattaneo et al. (1998) and Joy et al. (2006) for a situation at a magnetised planet. There, a prevalence of dips occurs deeper in the magnetosheath, suggesting that the structures around perihelion at the comet may have been created in the wake of the quasi-perpendicular shock.

    When the outgassing rate is lower, a weak asymmetric shock may form (that is, the first stage of the shock's formation, see

the 'infant bow shock´ as reported by Gunell et al., 2018; Goetz et al., 2021). However, such a shock-like structure may be too weak to generate the temperature anisotropy to drive mirror-mode unstable conditions in the plasma, in which case the pick-up ion process, always present and linked to the EUV flux, would become the leading driver of mirror-mode generation. This is likely the case during most of the Rosetta mission. Indeed, when the outgassing rate is even lower, no shock is expected to form, as would be the case in the early stages of the mission (up to about Spring 2015) or, equivalently, later in the mission

(after January 2016). However, mirror mode-like structures are still detected then. All of these effects are thus likely to be





mixed in Fig. 7, which may prevent any clear trends to be seen. Because of relatively low statistics for the number of structures found in the Rosetta dataset, we cannot investigate this further at this stage. A precise understanding of all of these aspects, combining global numerical simulations driven by inputs from in-situ observations, is left to a future study.

## 4   Discussion and Conclusions

In this study, we have used and adapted a well-known magnetic field-only method to identify mirror mode structures in the plasma environment of comet 67P.

A modification of the method was necessary, as the original implementation misidentified other compressional structures in the magnetic field as mirror mode events. The adapted method now takes into account the antiphase behaviour of the plasma density and the total magnetic field (which are expected characteristics of mirror modes), as well as the shape of the magnetic

field and plasma density. With this, we were able to identify over 500 mirror mode-like events.

The characteristics of the identified events are in general in accordance with previous studies, although events tend to be more isolated (a single wave packet) than expected (Volwerk et al., 2016a). Only one clear event containing a train of mirror modes was found. However, many events are embedded in an environment that has signatures of mirror modes that do not satisfy our stringent criteria, which is also expected from an automatic algorithm (see Simon Wedlund et al., 2022, for a critical

account). Mirror modes are expected to be mainly generated at comets due to the presence of unstable pick-up ion distributions or in the wake of the quasi-perpendicular bow shock. Since we find mirror modes even at outgassing rates that are too low for a bow shock to form, and there is no dependence of mirror mode occurrence on the cometary activity, we conclude that mirror modes at 67P are predominantly generated through a pick-up ion instability. Near perihelion, at high outgassing conditions, mirror modes could also be generated behind the bow shock far upstream of their place of detection, as their morphology is

consistent with mirror modes that are created at planetary bow shocks and convect downstream (Simon Wedlund et al., 2022). There is also evidence to suggest that mirror modes are generated upstream of the measurement point since the normalised length scales $L^*$ for the events described in this study are slightly larger than what was previously found for locally generated mirror modes (see section 3.1). As pick-up heavy ions are born upstream in the solar wind resulting in the classic ring-beam velocity distribution function, a temperature anisotropy may already arise in the solar wind plasma. Such a distribution function

can give rise to mirror mode waves and ion cylotron waves. No ion cyclotron waves were observed during the Rosetta mission, possibly because the plasma beta is higher and therefore mirror modes are preferentially generated. Mirror modes are then convected downstream with the plasma and diffuse or are possibly destroyed by plasma turbulence (Hasegawa and Tsurutani, 2011; Volwerk et al., 2008). Unfortunately, no reliable plasma temperature observations are available with Rosetta and an estimation of the plasma beta was not possible.

Mirror modes are found with a certain detection rate whenever data is available and there are no discernible trends with regards to gas production rate, magnetic field strength, or location in the coma. This is another indication that mirror modes are not generated locally, as the local plasma parameters and those in the generation region do not have to be the same.



In this study, we found that 10 out of 23 magnetic holes shown by Plaschke et al. (2018) were also identified as mirror modes by the selection method described in Sect. 2.2. This is unsurprising as mirror modes and magnetic holes share the characteristics that our detection method searches for; they are both compressional, pressure-balanced structures. Consequently, they are often thought to be related (Winterhalter et al., 1995). Plaschke et al. (2018) showed that magnetic holes are still observed in the inner coma, although solar wind protons are mostly replaced by cometary ions. This should also apply to mirror modes that are generated in the solar wind-dominated part of the cometosheath and convect into the cometary ion-dominated part.

In conclusion, we performed case studies and statistical studies of mirror modes in the cometary environment for the first time using the Rosetta datasets. Our findings indicate that the mirror mode-like structures we see are most likely generated non-locally through a pick-up ion instability instead of the more classical planetary mechanism of perpendicular acceleration due to quasi-perpendicular shock conditions. This is in keeping with the results from Giotto's historical flyby of comet 1P/Halley (Mazelle et al., 1991; Schmid et al., 2014).

As plasma density data was used to filter many misidentified mirror modes by the magnetic field-only method, we emphasize that for future cometary space missions, plasma density detectors with 1-to-2-s temporal resolution and a complete field of view, are mandatory in order to characterise and study mirror mode phenomena. Moreover, ion temperature measurements with good accuracy are needed to derive the plasma-$\beta$ parameter and temperature anisotropy, two essential ingredients for the generation of instabilities.

*Data availability.* All Rosetta data is freely available on the Planetary Science Archive, hosted by ESA http://psa.esa.int. The MAG data that was used was the most up to date version (V9.0) at the time. A full list of the mirror mode events identified in this paper may be found here: https://doi.org/10.5281/zenodo.7685489

## Appendix A: Mirror modes detection algorithm

In the following section, we dive into details about the selection method described at Sect. 2.2, namely, the anti-correlation (Fig. A1) and the peak/dip (Fig. A2) identification methods. For both figures, data were smoothed with a moving average over 3 s which allows for the removal of the highest frequencies since we are interested in mirror modes (low-frequency structures).

On the one hand, as is shown in Fig. A1, although both mirror mode candidates (panel a and b) satisfy the $B$–$n$ anti-correlation criterion, panel a shows a linearly shaped event instead of a wave-like one. Many of the mirror mode candidates we found exhibit this behaviour. Thus, it was necessary to filter out those false positive events in order to retain structures like that of panel b (see single event shown in Fig, 2, representative of these structures).

On the other hand, Fig. A2 explores how those linear events were filtered out. Panel a shows the expected mirror mode shape

with a clear $B$–$n$ anti-correlation and the simultaneous plasma density peak and magnetic field strength dip. Panel b shows a linearly shaped plasma density data (blue instead of purple line). This event does not exhibit simultaneity between the plasma density peak and the magnetic field strength dip. If any of these conditions were not satisfied, the event was dropped.

## Appendix B: Mirror modes morphology

To study the morphology of the mirror modes, for each previously identified event, we determine if the change in the magnetic

field magnitude is an increase (peak) or decrease (dip) above the background magnetic field. We found that some events exhibited both peaks and dips as shown in Fig. A3. Therefore, another category, labeled 'both', was also included in Fig. 7. Such a behaviour was also found in the study of Joy et al. (2006), where structures labelled as 'others' (similar to our 'both' category) were seen everywhere in the magnetosheath, with a maximum of occurrence closer to the bow shock. This behaviour is also seen at Earth (Soucek et al., 2008).

*Author contributions.* Conceptualisation and data analysis: ATF, CG. Writing: ATF, CG, CSW, MV. Editing and Discussion: all authors. Software and Visualization: ATF. Supervision: CG

*Competing interests.* None

*Acknowledgements.* The authors would like to acknowledge ISSI for the opportunity it offered for very valuable discussions on this topic as

part of the International Team # 499 "Similarities and Differences in the Plasma at Comets and Mars" and Team # 517 "Towards a Unifying





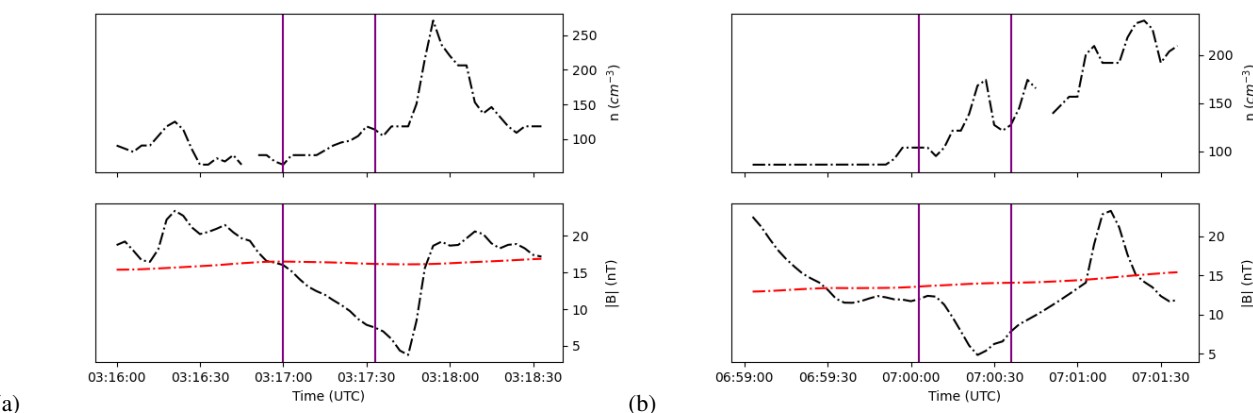

**Figure A1.** Examples of mirror mode candidates found on 25 May 2015 with $B$–$n$ anti-correlation. Plasma density and magnetic field strength in black and magnetic field strength's rolling average in red. **(a)** The mirror mode candidate is marked by purple lines lasting from 03:17:00 to 03:17:33 with $\mathcal{R}(N_{pl}, |B|) = -0.85$ **(b)** The mirror mode candidates are marked by purple lines lasting from 07:00:03 to 07:00:36 with $\mathcal{R}(N_{pl}, |B|) = -0.96$.

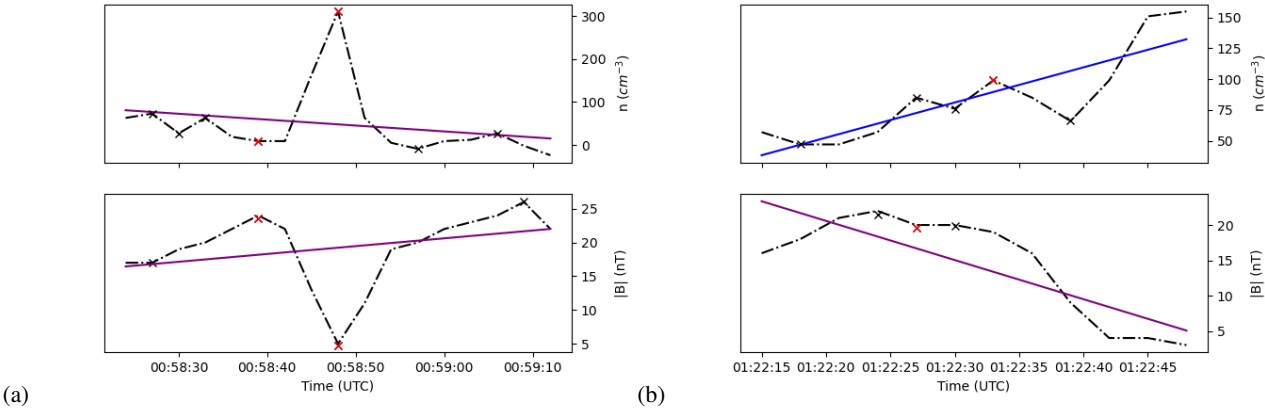

**Figure A2.** Peak identification using the prominence method for 8th and 14th February 2015 respectively. Black crosses represent the local extrema (local maxima and minima) while the red crosses represent the extrema with the highest prominence of the structures. In order to drop events without peaks or dips like Fig. A1 (a) linear fits were computed separately for the plasma density and magnetic field strength. If any of the $\mathcal{R}^2$ value is higher than 0.7 the event is dropped. Purple and blue lines show a satisfied and unsatisfied $\mathcal{R}^2$ criterion respectively for the linear fits, with **(a)** Lasting from 00:58:24 to 00:59:12 with $\mathcal{R}^2(N_{pl}) = 0.07$ and $\mathcal{R}^2(|B|) = 0.1$ **(b)** Lasting from 01:22:15 to 01:22:48 with $\mathcal{R}^2(N_{pl}) = 0.72$ and $\mathcal{R}^2(|B|) = 0.68$.

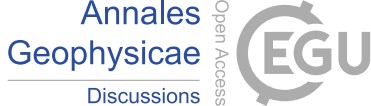

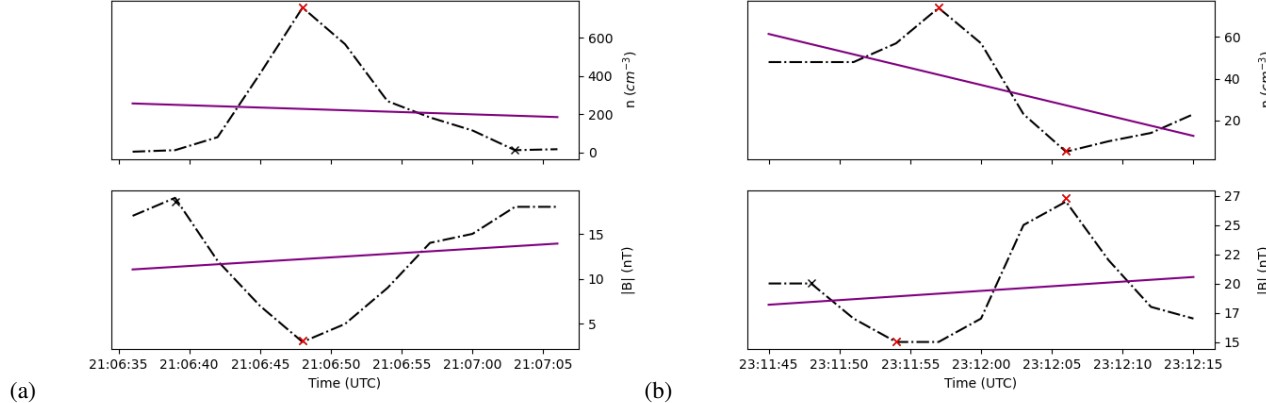

(a)                                                    (b)

**Figure A3.** Black crosses represent the local extrema (local maxima and minima) while the red crosses represent the extrema with the highest prominence of the structures. Purple lines show a satisfied $\mathcal{R}^2$ criterion for the linear fits. **(a)** Detection of a decrease in the magnetic field strength in antiphase with the plasma density on 18 February 2015. **(b)** Detection of both increase and decrease in the magnetic field strength in antiphase with the plasma density on 16 February 2015.

Model for Magnetic Depressions in Space Plasmas". CG was supported by an ESA Research Fellowship. ATF was supported by a Leiden University - ESA LEAPS project. The work by CSW is supported by the Austrian Science Fund (FWF) under projects N32035-N36 and P35954-N. AM was funded by the Swedish National Space Agency (SNSA) Grant 132/19.





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
