# Peer review of "Revisiting Mirror Modes in the Plasma Environment of Comet 67P/Churyumov-Gerasimenko"

_Annales Geophysicae, 2023_

## Author Response (AR1)

**Review of "Revisiting Mirror Modes in the Plasma Environment of Comet 67P/Churyumov-Gerasimenko" by Fallau, Goetz, Simon Wedlund, Volwerk and Moeslinger**

**Referee #1**

This is an interesting paper but it is unconvincing that it is correctly identifying solitary mirror mode waves. If one looks at Figure 1 and the identification of the mirror modes, the readership will ask what are all the other waves which have not been identified as mirror modes? They look pretty much the same as the events that you have identified. Is it just by chance that sometimes the angular changes fit their conditions that the authors have set for mirror modes and sometimes they do not? I realize that the authors are claiming that these other waves are evolutionary structures of mirror modes, but there has not been any past observations/evidence of this. This is just pure speculation. Maybe this is an entirely different wave mode, or possible this is a coupling of mirror modes with another wave mode? Or could there be two instabilities occurring, say a mirror mode instability and an ion cyclotron wave instability? The authors should discuss this in some detail. Give balance to the paper so that the readership will not be misled.

– We thank the reviewer for their input and have made changes to address the comments. The changes are highlighted in the attached file. There seems to be some misunderstanding of the main points of the paper, so we have reformulated to hopefully avoid any confusion.
The core of the method that we use is well known and has been used many times before to identify mirror modes. Of course the choice of parameters influences the method outcome, so we have either chosen values that other publications have used (for compatibility) or explained why an adjustment was necessary. This results in a catalog of mirror mode wave candidates that is the basis of a statistical study. This is an established method of investigation. We have discussed in the text why some events that could be mirror modes are not identified. Nowhere in the draft do we intend to claim that we present evidence of mirror modes developing out of/into magnetic holes. We merely point out their similarity and that there is a theory that they are related. We have reformulated the text to make this clearer to the reader. No ion cyclotron waves have been reported in the cometary environment of comet 67P and the aim of this paper is not to search for them.

In light of this I suggest that the authors put in the word "possible" in the title before "Mirror Modes" and in line 3 of the abstract. Future work on this by other people might be able to resolve what is creating these perplexing waves.

– The method that was used in this paper has been used many times to identify mirror mode structures. Mirror modes have been reported at comets. While the interpretation of the generation of the waves is certainly difficult, it is not warranted to discount the characterisation of the identified events as only "possible" mirror mode waves. In Sect. 2

we emphasize which waves other than mirror modes could be detected by the original B-field-only detection scheme and present several additions and ways of mitigating this (B-n anticorrelation, wave-like morphology): all of those observational additions, paired with theoretical and modelling results, are strongly compatible with the actual presence of mirror modes. We therefore opt to retain the title as it is, but changed the abstract's 3rd line to "mirror mode-like" to emphasize that, because the ion/electron temperature information was impossible to derive with accuracy, the origin of the waves we see is difficult to ascertain, as already stated at the end of Sect. 2.

In reading through the paper, it appears as if this is a reporting of the authors' efforts in research and not a streamlined scientific paper. In the abstract, sentence line 8 -9, you mention a magnetic field only method of detecting mirror modes and later in the body of the text, you discard it for a magnetic field and plasma density method.

– The magnetic-field only method was used as the first step of the multi-step identification process. It was not discarded, but added to. We have clarified this in the text.

It would be best if you delete this sentence in the abstract and in the body of the text and get to the main point of the results. In the sentence on lines 9-10, I think you should correctly state that the 565 events were detected by the standard technique of identifying mirror mode events (anticorrelated magnetic fields and plasma densities)? This is not a new idea and should be the starting point of your discussions.

– Previous studies just used the magnetic field only method to identify mirror modes in planetary or cometary environments. One of the realizations of this study is that this method is not enough to correctly identify mirror modes at comet 67P as stated in the text. As we detail the selection method, we also reduce the number of events, for repeatability and traceability we have added the number of events at each step.

Abstract, sentence lines 12-13, you mention that you have detected only one mirror mode train. This is interesting and should be expanded upon in the body of the text. It should be noted that for mirror modes in planetary magnetosheaths, the waves are always in trains, not single events separated by other types of waves. It should also be noted that previously observed mirror modes at comets were trains, but of very short duration. Some discussion of this needs to be added. References to mirror modes at comets are: GRL, 14, 644, 1987 doi:10.1016/S0237-1177(97)004730; JGR 98, 20955, 1993; NPG, 6, 229, 1999 doi:10.5194/npg-6-229-1999. These references should be added to the paper. Previous reports of mirror modes in planetary sheaths and at comets (and elsewhere) have all reported trains, not single events as shown here. This should be discussed in the body of the text.

– The lack of mirror mode trains is indeed interesting. For the abstract we have retained the text as is, but we have added a more detailed discussion in the main body of the text. The requested references were also added in the text.

Detailed Comments

Introduction line 27. I suggest the deletion of the Ip reference and the replacement by Wu and Davidson.  Ip was just a copy of the original.  They did nothing new.

 – Done

Lines 27-28, references should be given for the different wave modes mentioned. For magnetosonic waves, the original discovery was GRL, 13, 3, 259, 1986. The most recent work on this is the Ostaszewski et al article that you quote.  Both should be mentioned here.

 –We have added the suggested reference (Tsurutani & Smith) as well as Tsurutani et al 1987 . We also included mention and references to lower hybrid waves, ion cyclotron waves, and singing comet waves.

The term "Alfven wave" should be deleted here.  It is realized that theoretically Alfven waves are the low frequency analog to ion cyclotron waves, but detailed studies of Alfven waves indicate that they are not left hand polarized, but are arc polarized spherical waves.  Please see JGRSP 123, https://doi.org/10.1002/2017JA024203 for details.  Nothing like these waves were detected at comets, thus the suggested deletion. The readership will be confused.

–Done

For ion cyclotron waves, a good reference is JGR, 98, 20921, 1993. You might wish to add other wave modes that have been detected at comets (and references).https://doi.org/10.1029/93JA02583.

 – Done

Line 32.  I suggest the deletion of the term "magnetic holes".  There are many people that do not believe that magnetic holes are mirror modes.  Please note the references in Tsurutani et al., 2011 which indicates many references to mechanisms for magnetic hole generation which are not temperature or pressure anisotropy generated. If you discuss magnetic holes here or elsewhere, these references should be quoted as alternative explanations.

–We have deleted the mention of magnetic holes in this context and discuss the relation between magnetic holes and mirror mode waves in more detail later on.

There has been a problem with the analysis of "linear magnetic holes" as well.  It is the same problem as discussed with your waves.  Singular events were selected out of a string of events that did not fit linear holes.  What are the rest of the events?  Are they due to some other mechanism? The idea that they are the evolution of mirror modes is not convincing.  I don't believe anyone has shown a case where mirror modes have only partially evolved. Again this is pure conjecture that these are due to the evolution of mirror mode waves.  Others have suggested mechanisms for magnetic hole formation. These should be quoted and discussed.

–We have added the references and discuss this in more detail.

Line 33. The original theoretical paper and observational paper on mirror modes and proton anisotropies should be quoted here instead of what you have. They are Phys Fluids, 12, 2642, 1969 and JGR, 87, A8, 6060, 1982.

–Done

Line 49. Remove "Alfven wave" here. See previous comment.

–Done

Line 51. Remove "Alfven" in front of ion cyclotron waves. See above comment. A couple of references on ion cyclotron waves should be added here: Remya et al. JGR, 118, 785, 2013 doi:10.1002/jgra.50091 and Remya et al. ApJ, 793:6, 2014 doi:10.1088/0004-637X/793/1/6. They are follow- ons to the Gary 1992 paper and give further information for the instability and the waves.

–We have added the publications by Remya et al. Ion cyclotron waves are still mentioned as the low beta mode.

Line 51, sentence beginning with "Moreover…". This sentence should be reworded. This was a result from theory but have never been verified experimentally. And a "minute" amount of heavy ions would not have much of an effect. Almost nothing depending on the "minuteness".

–We have rephrased the sentence.

Line 52. Remove "Alfven"

–Done

Line 60. A reference to field-line draping should be added here. I recommend the original paper: JGR 68, 5111, 1963.

–Added

Line 71. Does the "two different sizes of mirror mode waves" still hold after you found that the magnetic field only technique produced errors? If not, then this sentence should be removed.

– The magnetic field only technique does not produce errors, as stated later on in the manuscript, the dataset has been improved since the original study which results in different identifications of mirror mode candidates in this study. Therefore this paper cannot make any statements about the size of the mirror modes found in the original study. As this is simply a statement of what was found previously, we do not see a need to remove it.

Lines 80-89.  Are all of your possible mirror modes pressure balance structures or can you not say?  This should be discussed in the paper.  On line 89 besides the Hasegawa (1969) reference, you should add Price et al.  1986 and Tsurutani et al. 1982.

–In this section we summarize the Volwerk 2016 study, the paragraphs are not related to this study. As our events are constrained to intervals where high resolution plasma density is available,  we only include events that are pressure balanced structures.

Lines 90-96, linear magnetic holes.  A comment concerning this was mentioned before.  Although this conjecture is possible, for balance some statement indicating that they are possibly not mirror mode structures should be made.  The many possible mechanisms for their generation should be quoted.

–See answer above.

Lines 96-99.  This conclusion is too strong.  These structures may not be mirror modes at all and may be generated by processes not involving anisotropic protons.  A statement should be added stating this.

– With the additions to the discussions on the origins of mirror modes and bearing in mind that this section simply refers to the results of other publications, we think this wording is warranted. It is not a conclusion at all, simply a summary of what others have found.

Lines 144-146.  It seems unnecessary to describe your selection process as a two step procedure.  Why don't you cut to the chase and state your final selection criteria?

–The selection method is a multi-step procedure and we describe it as such.

Table 1.  Your criteria for detection of your events are clear.  But later in the paper you mention that some of the selected events do not have anticorrelations between B and N.  This seems contradictory.  Can you please clean this up?  Throw out events that don't have clear anticorrelations?

–Our final event list does not contain any events without a clear anticorrelation.

Lines 171-176.  Is all this detail necessary to tell the readership of AG?  I think it is important to tell the readership that the "magnetic field only technique" does not work for identifying possible mirror mode waves at your specific comet, but I think this could be done in one sentence. When you mention different wave types earlier in the paper, you should add "steepened waves" (plus reference) and "singing comet waves" (and references).

– We have added references as requested. The detail in the description  is necessary to inform the reader who may want to do a study similar to this one, that the magnetic field only method misidentifies a well known wave mode as a mirror mode because it disregards the pressure balance nature of the waves. This is why we inform the reader of the problem and the solution (add anti-correlation)

Line 181. Telling the reader that the Volwerk et al. (2016a) results are now known to be incorrect will be useful to the readership, but perhaps state this more succinctly?

–We have reformulated this section.

Line 185. Why don't you start the true data analysis section with the discussion of the Wedlund et al. (2022) method which works? Why discuss methods which don't work? And by the way this is not a new method but was discussed by Hasegawa 1969; Tsurutani et al. 1982 and Price et al. 1986. Plus of course the anisotropic protons and pressure balance, both topics which you have not discussed/shown. This latter point should be mentioned prominently. Since you were not able to do these types of analyses, your method only partially identifies mirror mode structures.

–We are of course aware that the usage of the magnetic field only method and the anti-correlation is not new. The text does not claim this. However, some additions were made to accommodate for the complexity of the cometary environment (peak identification method). We have made an attempt to reformulate this description slightly to better convey the points.

Line 205. What does the presence of magnetic holes have to do with the generation of mirror modes? Please explain or delete. Also note previous comments that many people do not believe that magnetic holes are the remnants of mirror modes. So this point should also be made if you wish to revise this sentence.

– This section is indeed misplaced. We simply meant to point out that temperature data is not available so we could not do any checks of the plasma beta or the anisotropy. We have therefore removed this paragraph and added a sentence on the unavailability of the plasma temperature earlier in the section.

Figure 1/lines 223-226. Please shade the intervals that you believe are mirror modes for the readership. I don't see where your criteria are being met.

–We have added shading to Figure 1 and 2.

Lines 227-233. Since these "events" do not meet your criteria, why don't you delete the vertical lines and delete these sentences?

–I think there was a misunderstanding here. All of the four intervals meet all criteria. Only the wave structures between them do not. We have shaded the four events and reformulated the text to make this clearer.

Lines 277-280. This is a highly biased statement. Another possibility is that magnetic holes have nothing to do with mirror modes. As previously mentioned there have been many proposed mechanisms for magnetic holes, which you should include in your revision. Please revise the discussion so it is more even handed for the readership of AG.

–We have reformulated this. We simply meant to point out that it has been shown before that plasma structures formed upstream in the solar wind or cometosheath can convect into the inner coma. As they do so, they are modified. This example holds whether or not magnetic holes and mirror modes are related.

Figure 3. I think this is perhaps the most interesting figure of the paper. It will help others who wish to tackle this wave problem to realize that this may have nothing to do with mirror mode generation.

–We are pleased that the referee finds this figure interesting, but we do not understand what the remainder of the comment is referring to.

Lines 328-342 discussion. The authors should mention that the previous studies of the Saturnian magnetosheath have shown no evolution of mirror modes into magnetic holes. Thus the comment by Plaschke et al 2018 is conjecture, not observations.

–As we think is clear in this paragraph, we do not talk about magnetic holes at Saturn. Later, we make a tentative comparison of a mirror mode deformation on its path downstream with the solar wind flow with the evolutionary morphology of a MH crossing the cometary interaction region as described by Plaschke+ 2018 (see their Fig. 5, which is a simplified sketch and example of possible deformation of a large-scale magnetic hole). We added the reference to Fig. 5 of Plaschke to help the reader understand what specific aspect we are referring to here. As discussed by Plaschke+ 2018, expected differences between a Jupiter/Saturn/Earth-like environment and cometary/Mercury-like environments has to do with scales of mirror mode-like structures as compared to the size of the interaction region (typically, the bow shock).

**Referee #2**

This article revisits the presence of mirror modes in the plasma environment of the comet 67P with Rosetta data. If the findings are not completely conclusive, the methods and discussions are worth reading and nicely complement previous studies on mirror modes. I have mostly minor comments to this version.

We thank the reviewer for their constructive comments and address them below. Proposed changes to the manuscript have been highlighted in the attached pdf.

l52: the peculiarities due to the presence of heavier ions are more general and not restricted "to the solar wind".

–We have substituted "to the solar wind" by "to the plasma"

l61-62: do the authors mean that the field line draping at the MP acts to increase the anisotropy ? Does this come in competition with the reduction of the anisotropy under the action of mirror mode activity itself (the instability "consumes" the free energy contained in the anisotropy) ? How does this competition take place ?

–Yes, the field line draping at the MP increases the anisotropy of the electron temperature. This is one of the mechanisms that was invoked by Tsurutani et al 2011 as a generator of mirror modes. There is no discussion in literature of how this mechanism competes with the reduction of anisotropy through the mirror modes themselves.

l95-96: on the theoretical possibility to observe mirror modes (mostly dips) in mirror stable plasmas, the authors could refer to Passot et al., 2006 (https://doi.org/10.1063/1.2356485) together with Génot et al., 2011 (https://doi.org/10.5194/angeo-29-1849-2011) which presents a scenario of mirror mode evolution based on simulation and observations. This also applies to the discussion around l330.

–Both papers are very good suggestions and we have added them in the discussion on the peaks and dips.

l120: same sentence as l143

– This was deleted.

l179: is there a way to check file versions on the Rosetta database ? (at ESAC ?)

–To our knowledge only version 9.0 (the most up to date version used here) and version 6.0 are archived on the PSA. Version 6.0 was published in May 2018 and the version used in the Volwerk et al publication is not archived.

l204: if no IC waves have been detected at 67P, why putting so stringent constrains on mirror mode detection ? ie, a reduction of 32000 to 565 events. What other mode could this be ?

–Fast magnetosonic waves are also of a compressional nature. Usually, in the solar wind for example, they have small amplitudes and therefore are not identified by the algorithm that mandates large amplitude fluctuations. However, in the cometary environment there are fast magnetosonic waves with a large amplitude, so called steepened waves. We added mention of steepened waves in the introduction to make the reader aware of their existence.

l380: throughout the paper the term "magnetic-field only method" is used. Why is it well-known ? Also the present paper complements the analysis with a check of the B-N anti-correlation. So I don't understand why insisting on the ""magnetic-field only". This comment agrees with the one of another referee who recommends a better description of the method used in the paper (I think mostly the naming should be adapted).

– It is well-known since it has been used in previous studies. "magnetic-field only" is named in literature, eg Volwerk et al (2016) and Simon Wedlund et al (2022, 2023). We have made modifications to the text to explain this better.

l385: the down selection from 32000 to 500 could be repeated here.

– Done

l387: the very rare observation of mirror mode trains is very puzzling, if not strange. Could this be linked to the method itself ? Otherwise I agree with the conclusion that most of the observed events are linked to mirror activity happening elsewhere and are just remnants of this.

– The method by itself just identifies single events for each second of data (B-field and plasma data alike, when the latter is available). The time difference between the identified 565 mirror modes was too high (hours) in order to be considered as trains. The identified mirror mode train was the only one with 4 single events identified at a relatively close time (less than 15 minutes of difference between events). We agree that this is puzzling and unexpected, but we do not see how this could be related to the method, as a very similar method applied at Mars (Simon Wedlund et al. 2022) does identify wave trains.

l414-415: to my knowledge, mirror modes in cometary environments have already been studied. What is done for the first time exactly ?

– This is the first time that mirror modes in the cometary environment could be studied over a long period of time and at different gas production rates. This allows for a statistical treatment of events and to relate the occurrence to parameters such as gas production, background magnetic field and position of the spacecraft within the environment.

We have added this to the text.

last sentence: sure it would be good to have such measurements. But what exactly remains to be understood ? And what do we learn, from the mirror activity, on the comet itself and/or cometary processes in general ?

– Many questions on the generation and evolution of the mirror mode in the plasma environment of 67P remain. In more general terms, this addresses the question of how energy is transferred between the solar wind and the cometary ions. We have added more detail to this last section.

---

## Author Response (AR2)

**Review of "Revisiting Mirror Modes in the Plasma Environment of Comet 67P/Churyumov-Gerasimenko" by Fallau, Goetz, Simon Wedlund, Volwerk and Moeslinger**

**Referee #1**

Referee second comment: I am dissatisfied with the responses of the authors in this section of the correspondence. I will give some further arguments below.
* * *
Referee first comment. This is an interesting paper but it is unconvincing that it is correctly identifying solitary mirror mode waves. If one looks at Figure 1 and the identification of the mirror modes, the readership will ask what are all the other waves which have not been identified as mirror modes? They look pretty much the same as the events that you have identified. Is it just by chance that sometimes the angular changes fit their conditions that the authors have set for mirror modes and sometimes they do not? I realize that the authors are claiming that these other waves are evolutionary structures of mirror modes, but there has not been any past observations/evidence of this. This is just pure speculation. Maybe this is an entirely different wave mode, or possible this is a coupling of mirror modes with another wave mode? Or could there be two instabilities occurring, say a mirror mode instability and an ion cyclotron wave instability? The authors should discuss this in some detail. Give balance to the paper so that the readership will not be misled.

Authors first response– We thank the reviewer for their input and have made changes to address the comments. The changes are highlighted in the attached file. There seems to be some misunderstanding of the main points of the paper, so we have reformulated to hopefully avoid any confusion. The core of the method that we use is well known and has been used many times before to identify mirror modes. Of course the choice of parameters influences the method outcome, so we have either chosen values that other publications have used (for compatibility) or explained why an adjustment was necessary. This results in a catalog of mirror mode wave candidates which is the basis of a statistical study. This is an established method of investigation. We have discussed in the text why some events that could be mirror modes are not identified. Nowhere in the draft do we intend to claim that we present evidence of mirror modes developing out of/into magnetic holes. We merely point out their similarity and that there is a theory that they are related. We have reformulated the text to make this clearer to the reader. No ion cyclotron waves have been reported in the cometary environment of comet 67P and the aim of this paper is not to search for them.

Second referee comment. The changes that you have made are insufficient. Every paper should be a stand alone work. Just because other papers have come up with similar conclusions doesn't mean that the present work is correct. All works are possibly in error. As one example when there were many potential errors in the literature concerning magnetic holes, an editor of JGR asked for a review paper straightening out the differences between magnetic holes and mirror mode waves. The published paper is JGR, 116, A02103, 2011. Doi:10.1029/2010JA015913. It seems to me that the only difference between the previous magnetic hole issue and this paper is that you are selecting single event "mirror modes" among other structures. What I am asking for is for you to explain to the readership what you think these other structures are which are not mirror modes.

As originally suggested, please give your opinion (to the readership) your opinion of what the other waves may or may not be. I do not believe you addressed this issue at all.
* * *
[Figure]

Second Author comment:
We shall try to address the referees concerns one by one below:
Solitary mirror modes are not a criterion, the method identifies individual mirror modes but it is not restricted to isolated events as is shown in Figure 1. Nowhere within the method do we require that these structures are solitary, this is something that results from the method.
As we mentioned in a previous response the selection method was defined as it is, in order to retain unambiguous mirror mode events, in the paper we mentioned that the rest of the events in Figure 1 could be also possible mirror modes but because of our stringent criteria, they were not identified by the method. As stated in the Tsurutani et al 2011 paper, the deltaB/B criteria are chosen somewhat arbitrarily. Our choice of criterion is based on what other authors have successfully used to identify MMs. We speculate that if we chose a slightly less restrictive criterion the structures within this wavetrain may also be identified as mirror mode-like. Specifically in the above figure we have marked the specific reason why those events were not identified as mirror modes by the algorithm. We included in the draft the following sentence "For example, the events closer to 00:12 and 00:17 UT were not identified as a mirror mode since they did not satisfy the theta and phi angles criterion."

Referees first comment. In light of this I suggest that the authors put in the word "possible" in the title before "Mirror Modes" and in line 3 of the abstract. Future work on this by other people might be able to resolve what is creating these perplexing waves.

First reply of authors– The method that was used in this paper has been used many times to identify mirror mode structures. Mirror modes have been reported at comets. While the interpretation of the generation of the waves is certainly difficult, it is not warranted to discount the characterisation of the identified events as only "possible" mirror mode waves. In Sect. 2 we emphasize which waves other than mirror modes could be detected by the original B-field-only detection scheme and present several additions and ways of mitigating this (B-n anticorrelation, wave-like morphology): all of those observational additions, paired with theoretical and modelling results, are strongly compatible with the actual presence of mirror modes. We therefore opt to

retain the title as it is, but changed the abstract's 3rd line to "mirror mode-like" to emphasize that, because the ion/electron temperature information was impossible to derive with accuracy, the origin of the waves we see is difficult to ascertain, as already stated at the end of Sect. 2.

Referee second comment. These structures that you are presenting as mirror modes are not like the continuous string of mirror modes that have been observed in planetary magnetosheaths and were discussed in 2011 JGR paper mentioned previously. Yes, mirror modes have previously been reported to occur in other places such as at comets and in interplanetary space. Those events were noted to be of continuous cycles of mirror mode structures.

I repeat, the word "possible" should be added to the title. I think this is a generous amendment.

On the non-detection of mirror mode trains:
It is not because we detect only a single structure at a time that these are not part of a train of mirror modes. Detection is not reality, it is only a tool. We do not want to include too many structures at the expense of having a lot of false positives, hence the strong criteria we chose.
The referee implies that if the MMs that we found are not trains then they are not mirror modes. This is clearly against previous studies, see e.g. Pokhotelov et al 2008 and Volwerk et al 2016.
Very often, mirror mode trains are identified using the magnetic field only method by single events and then the entire interval is classified as a mirror mode train (e.g. Simon Wedlund et al 2023). Our results are similar. MM trains are seldom captured in one fell swoop by any automatic algorithms. We are actually more stringent here, and do not claim that the events in figure 1 are all part of a mirror mode train.
Other papers that only use the magnetic field only method (which is inferior to our method here) also claim the name mirror mode in the title without any modifiers. We do not see how this case would be different (Erdos 1996, Lucek 1999, Volwerk2016, Volwerk2021).
* * *
Referee first comment. In reading through the paper, it appears as if this is a reporting of the authors' efforts in research and not a streamlined scientific paper. In the abstract, sentence line 8 -9, you mention a magnetic field only method of detecting mirror modes and later in the body of the text, you discard it for a magnetic field and plasma density method.

Authors first reply– The magnetic-field only method was used as the first step of the multi-step identification process. It was not discarded, but added to. We have clarified this in the text.

Referee first comment. It would be best if you delete this sentence in the abstract and in the body of the text and get to the main point of the results. In the sentence on lines 9-10, I think you should correctly state that the 565 events were detected by the standard technique of id entifying mirror mode events (anticorrelated magnetic fields and plasma densities)? This is not a new idea and should be the starting point of your discussions.

Authors first reply. Previous studies just used the magnetic field only method to identify mirror modes in planetary or cometary environments. One of the realizations of this study is that this method is not enough to correctly identify mirror modes at comet 67P as stated in the text. As we detail the selection method, we also reduce the number of events, for repeatability and traceability we have added the number of events at each step.

Referee second comment. I disagree with this statement. The method to identify mirror mode structures was published in JGR, 87, A8, 6060-6072, 1982. It was pressure balance structures with little or no magnetic field deviations. The magnetic field alone technique was proposed as a shorter analytical method, which you now find does not work. You are going back to the original method except not even fulfilling all of the diagnostic techniques. Please give a correct and much shortened description of your criteria. For example you could say that the original 1982 technique

was use but because you do not have pressure balance information you have forgone that information? That basically covers your final analytical method.

Second author response: We reiterate that the magnetic field only method may work well in other environments but does not work in the cometary environment.
Clearly a number of methods have been proposed to identify mirror modes. All of them have drawbacks. The method we have chosen (the entire process, not just the magnetic field only step) is based on the methods recently used at comet 67P and then refined to take into account the specifics of the cometary environment and the availability of data products. In no way do we claim that the method is perfect; indeed we do the opposite and clearly state the assumptions and limitations of the method.
Quite frankly, we do not understand the referees argument here as the method they suggest we look at is one that is also flawed, and not possible to use given that we have no temperature or plasma pressure measurements at the comet and only one spacecraft. We have included a very thorough description of the methodology so as to ensure proper scientific method through reproducibility.

Topic editor's comments

Regarding specific comments:

Referee 1 ---"As originally suggested, please give your opinion (to the readership) your opinion of what the other waves may or may not be. I do not believe you addressed this issue at all."
---I believe this to be a reasonable request since the text around line 240 (for example, but not limited to) does raise a number of questions, and probably should be placed on a more scientifically solid footing. e.g. the sentence "We suspect ... but could still be ... " does suggest things that you probably don't want it to suggest?
("Although only the four marked intervals were identified as unambiguous mirror mode structures, there are other signatures 240 in the magnetic field, especially between 00:14 and 00:17 UT, that resemble the other mirror mode events. We suspect that these do not fulfil our stringent criteria but could still be mirror modes, following similar remarks made in Simon Wedlund et al. (2022). This is the only interval in our database where such a train of mirror modes was observed.")

Referee 1 ---"I repeat, the word "possible" should be added to the title. I think this is a generous amendment."
---I am inclined to sympathise with the referee here. If the authors have decided to change the text in the abstract to "mirror mode-like" in then a similar qualifier should also be in the title? I also note that discussions regarding mirror-mode and mirror-mode-like appear in other sections and suggest different levels of ambiguity at different points. It seems reasonable to modify the title in some way that acknowledges the variety of wave modes discussed in this paper, and the detailed discussions. And I don't think this is a weakness, rather it is a strength.
Referee 1 ---"I disagree with this statement. The method to identify mirror mode structures was published in JGR, 87, A8, 6060-6072, 1982. It was pressure balance structures with little or no magnetic field deviations. The magnetic field alone technique was proposed as a shorter analytical method, which you now find does not work. You are going back to the original method except not even fulfilling all of the diagnostic techniques. Please give a correct and much shortened description of your criteria. For example you could say that the original 1982 technique was use but because you do not have pressure balance information you have forgone that information? That basically covers your final analytical method."
--- I should imagine you can find a way to accommodate this request relatively easily, one way or another?
Of course, please do offer a respectful rebuttal if you have strong views in the other direction(s).

Unfortunately, we strongly disagree with the points made by the referee. We have outlined our concerns in detail above. Some of the referees statements in this round clearly ignored additions made to the paper in the last round, so we have reiterated those.

We also have trouble understanding why the referee insists on changing the title, when many papers have been published, using less stringent methods, and made no mention of the word "possible" in the title. In the text we state clearly that, as with any method, we cannot be 100% sure that all the structures we find are mirror modes, but we do not know of any other structures that fulfill all of these criteria (Wedlund et al. 2023). This is why the criteria were chosen to be quite stringent, so as to reduce the number of false positives.

The method described by Tsurutani et al 1982 cannot be applied to the data from Rosetta at the comet. Other well-documented and discussed methods are feasible and we have chosen to use them. Does the referee imply that only the original 1982 method is correct and all publications since then have been wrong? While the referee may believe this, many other authors, including the second referee would disagree.
If we shorten the description of our method, our results would not be reproducible and we believe that this is not in line with a proper scientific paper.